# Do age, gender, and education modify the effectiveness of app-delivered and tailored self-management support among adults with low back pain?—Secondary analysis of the SELFBACK randomised controlled trial

**Ellen Marie Bardal**[1,2‡]*, **Louise Fleng Sandal**[3‡], **Tom Ivar Lund Nilsen**[4,5], **Barbara I. Nicholl**[6], **Paul Jarle Mork**[4‡], **Karen Søgaard**[3‡]

**1** Department of Neuromedicine and Movement Science, Norwegian University of Science and Technology (NTNU), Trondheim, Norway, **2** Clinic of Rehabilitation, St Olavs Hospital, Trondheim University Hospital, Trondheim, Norway, **3** Department of Sports Science and Clinical Biomechanics, University of Southern Denmark (UoSD), Odense M, Denmark, **4** Department of Public Health and Nursing, Norwegian University of Science and Technology (NTNU), Trondheim, Norway, **5** Clinic of Anaesthesia and Intensive Care, St Olavs Hospital, Trondheim University Hospital, Trondheim, Norway, **6** Institute of Health and Wellbeing, University of Glasgow (GLA), Glasgow, United Kingdom

‡ EMB and LFS share first authorship on this work. PJM and KS share last authorship on this work.
* ellen.bardal@ntnu.no

**Data Availability Statement:** In line with the ethics regulations in Norway and the obligations related to

## Abstract

SELFBACK is an artificial intelligence based self-management app for low back pain (LBP) recently reported to reduce LBP-related disability. The aim of this study was to examine if age, gender, or education modify the effectiveness of the SELFBACK intervention using secondary analysis of the SELFBACK randomized controlled trial. Persons seeking care for LBP were recruited from primary care in Denmark and Norway and an outpatient clinic (Denmark). The intervention group (n = 232) received the SELFBACK app adjunct to usual care. The control group (n = 229) received usual care only. Analyses were stratified by age (18–34, 35–64, ≥65 years), gender (male, female), and education (≤12, >12 years) to investigate differences in effect at three and nine months follow-up on LBP-related disability (Roland-Morris Disability Questionnaire [RMDQ]), LBP intensity and pain self-efficacy. Overall, there was no effect modification for any of the sociodemographic factors. However, data on LBP-related disability suggest that the effect of the intervention was somewhat more beneficial in older than in younger participants. The difference between the intervention and control group due to interaction was 2.6 (95% CI: 0.4 to 4.9) RMDQ points for those aged ≥65 years as compared to those aged 35–64 years. In conclusion, age, gender, or education did not influence the effect of the SELFBACK intervention on LBP-related disability. However, older participants may have an additional long-term positive effect compared to younger participants.

**Trial registration**: ClinicalTrials.gov Identifier: NCT03798288.

the EU financed selfBACK project, the data will be anonymized and made publicly available after a 5-year period after the completion of the study (i.e., March 2026). During this 5-year period a data access committee will ensure availability of the data underlying the results presented in the study by request (email: ism-post@mh.ntnu.no). Researchers, who propose further use of selfBACK data, may also contact the steering committee of the selfBACK project (contact to Paul Jarle Mork paul.mork@ntnu.no or Karen Søgaard) for approval of use.

**Funding:** The SELFBACK project has received funding from the European Union Horizon 2020 research and innovation program (grant agreement no 689043 to PJM). The funders had no role in study design, data collection and analysis, decision to publish, or preparation of the manuscript.

**Competing interests:** The authors have declared that no competing interests exist.

## Author summary

Supporting people to self-manage their low back pain (LBP) is an important aspect of treatment. Delivery of personalized or tailored support via smartphone apps has shown promising results in reducing both pain and disability in people experiencing LBP. These apps have been mostly tested on middle-aged and well-educated women and we do not know if other groups have the same benefits. To be able to recommend self-management apps for people with LBP, it is important to identify sub-groups that may experience particularly favorable effects of such interventions, or alternatively, sub-groups without benefit or who may experience worse outcomes. In this study we wanted to examine if the effect of tailored support via a smartphone app (SELFBACK) on LBP related disability was modified by age, gender and education. Our analysis showed that age, gender, or education did not impact the effect of the SELFBACK intervention. However, older participants may have an additional long-term positive effect compared to younger participants. This suggests that the SELFBACK intervention may benefit all persons seeking care for LBP in primary care regardless of their age, gender, and education, and be a helpful tool for clinicians and patients to support self-management of LBP.

## Introduction

Low back pain (LBP) is the leading cause of long-term disability worldwide [1] and one of the most common reasons for consulting a primary care physician [2,3]. LBP is prevalent across the entire lifespan and increases with age [1]. The high prevalence of LBP translates into a substantial personal burden for people experiencing LBP and a financial burden for society, including both welfare and health care systems [1,4].

Clinical guidelines recommend tailored self-management support as a core component of LBP treatment [5–7]. Self-management programs often target behavioral factors and aim to empower people experiencing LBP to manage their symptoms and make choices that may improve their wellbeing, function and pain level [8]. A recent meta-analysis showed that self-management support delivered via mHealth applications (e.g., smartphone apps) may reduce pain and disability in patients with LBP [9]. Smartphone apps may therefore be a viable option to reinforce, facilitate and improve tailored self-management of LBP. The SELFBACK app is an artificial intelligence (AI) based self-management app for LBP that provides evidence-based content tailored to the needs and characteristics of individual patients [10]. We have recently reported on the effectiveness of the app, showing that people receiving the SELFBACK app adjunct to usual care experienced reduced LBP-related disability and LBP intensity compared to those receiving usual care only [11], however, the effect was small and of uncertain clinical significance. Further, different factors that might modify the effect of the app have been explored. Neither stress or symptoms of depression [12], nor multimorbidity and co-occurring musculoskeletal pain [13], was shown to modify the effect of the SELFBACK app on LBP-related disability. The participants in the SELFBACK randomised controlled trial (RCT) form a heterogenous group in terms of age, gender, and educational background, which strengthens the generalizability of the results. Trials investigating digital self-management support for LBP have mainly included middle-aged [9,14] and well-educated females [14]. Currently, it is unclear if factors such as age, gender, and educational status modify the effect of digital self-management support for LBP. Older adults represent a large potential user-group of digital self-management tools and the proportion of older adults who own smartphones is rising [15].

A qualitative study among older adults with chronic pain found them willing and interested in utilizing mHealth solutions to manage their pain problems [16].

It has been proposed that people with higher educational levels are capable of managing their LBP [17,18] and may perceive mHealth solutions as more useful than people with lower educational levels [19]. However, neither education nor gender have previously been investigated as effect modifiers in digital or mHealth interventions in LBP. Investigating potential effect modifier may help to identify sub-groups that experience particularly favorable effects of an intervention, or alternatively, sub-groups without benefit or who experience worse outcomes [20]. The aim of the present study was therefore to examine if the effect of the sᴇʟꜰBACK intervention on LBP-related-disability, LBP intensity and pain self-efficacy is modified by age, gender, or education.

## Methods

The present study is an exploratory, secondary sub-group analysis of the sᴇʟꜰBACK RCT. The RCT was registered in ClinicalTrials.gov prior to trial start (NCT03798288, January 7, 2019) and the trial followed a predefined protocol [21]. The RCT is reported following the CONSORT guideline [22] and eHealth extension [23]. The study protocol was approved by the data protection agency in Denmark (201-57-0008) and the regional committees for ethics in medical research in Norway (2017/923-6) and Denmark (S-20182000-24). All participants gave their written informed consent before entering the RCT.

### Participants

Persons seeking care for LBP were recruited into the study by primary care clinicians (i.e., physiotherapists, chiropractors, or general practitioners) in Denmark and Norway and from an outpatient clinic for back pain in Southern Denmark. Inclusion criteria were: LBP within the preceding 8 weeks; a score of 6 points or higher on the Roland-Morris Disability Questionnaire (RMDQ) [24]; owning a smartphone with an iOS or Android operating system; a working email address. Exclusion criteria were: <18 years of age; inability to understand, speak or read the national language; mental or physical conditions limiting participation; unable to exercise; diagnosed with fibromyalgia; current pregnancy; previous spinal surgery; current participation in other LBP research projects.

Participants were randomized to the intervention group (sᴇʟꜰBACK in addition to usual care) or the control group (usual care only) in a 1:1 allocation ratio using permuted blocks with random sizes from 4 to 20. Randomisation was stratified by country (Denmark or Norway) and type of care provider (physiotherapist, chiropractor, general practitioner, outpatient clinic).

### Interventions

Participants in the control group were instructed to follow the advice and treatment given by their clinician. The intervention group had access to the sᴇʟꜰBACK app in addition to following the advice and treatment given by their clinician. The participants were informed that the app was just a supplement to their usual care and that they should follow any advice given by their clinician. Participants in the intervention group downloaded the sᴇʟꜰBACK app on their smartphone and synchronized the app to a step-detecting wristband (Mi Band3, Xiaomi) under supervision of a research assistant, who gave a general introduction to the content and use of the app. The sᴇʟꜰBACK app offered weekly, individually tailored self-management plans targeting physical activity, strength and flexibility exercises, and education. Detailed information about the app is published elsewhere [10,11,25,26].

## Effect modifiers

The effect modifiers investigated in this study are age, gender and education collected in the baseline questionnaire. Participants reported their age (number of years), gender (male, female) and their educational background (less than 10 years of schooling, 10 to 12 years of schooling, or more than 12 years of schooling). To assess a potential effect modification, the population were stratified based on their baseline age (18–34, 35–64, $\geq$65 years), gender (male, female) and education length ($\leq$12, >12 years), respectively.

## Outcomes

Participants answered a web-based questionnaire at inclusion (baseline) and at six weeks, three, six-, and nine months follow-up. All outcome variables collected in the SELFBACK RCT are presented in the published protocol [21]. The primary outcome of the RCT was LBP-related disability at three months follow-up assessed by the RMDQ [24]. RMDQ ranges from zero to 24 points with higher scores indicating more LBP-related disability. RMDQ is a recommended outcome for clinical trials in non-specific LBP [27]. Function is strongly related to both pain and psychosocial attributes [28], and two of the predefined secondary outcomes, LBP intensity and pain self-efficacy were therefore included. LBP intensity was assessed as average LBP within the past week on an 11-point numerical rating scale (NRS) ranging from zero to 10 [29]. Confidence in ability to cope despite pain was assessed by the Pain Self-Efficacy Questionnaire (PSEQ; range, 0–60; higher scores indicating greater confidence) [30].

## Statistical analysis

This study utilises the data from the 461 participants in the SELFBACK RCT [11]. The sample size calculation for the RCT has been described in detailed elsewhere [21]. We investigated differences in effect between the intervention and control group within the specified strata of age, gender, and education at three and nine months using all available time points as a priori described in the statistical analyses plan for the SELFBACK RCT (point 6.2); see supplement 2 in [11]. The analyses followed an intention-to-treat principle. We estimated the mean group difference in outcomes with 95% confidence intervals (CIs) for the intervention and control group from a mixed linear model using a constrained longitudinal analysis approach with a common baseline to gain efficiency and control for any baseline imbalances in the outcome. All estimated mean differences were adjusted for the variables used for the stratified randomisation of the RCT (i.e., country, care provider) and potentially important prognostic factors (age [years], gender [male, female], education [<10, 10–12, >12 years of education], duration of LBP [<1, 1–4, 5–12, >12 weeks] and average LBP intensity past week [11-point NRS]). In models stratified by either age, gender or education, the respective stratifying variable was not adjusted for. Assumptions related to normality and homogeneity of residuals, and normality of random intercepts, were assessed for all models. Based on the constrained longitudinal analyses, estimates of effect modification due to interaction were calculated as the difference in effect across stratum at three and nine months follow-up [20].

## Results

In total, 461 participants were randomized to the control group (n = 229) or the intervention group (n = 232). Table 1 shows the characteristics of the participants stratified according to age, gender, and education. S1 Table provides an overview of population characteristics within the intervention and control group, also stratified by age, gender, and education. The response

**Table 1. Population characteristics across sociodemographic variables.**

| | Age | | | Gender | | Education | |
|---|---|---|---|---|---|---|---|
| | 18–34 years (n = 103) | 35–64 years (n = 295) | 65+ years (n = 63) | Male (n = 206) | Female (n = 255) | ≤12 years (n = 164) | >12 years (n = 297) |
| Country of recruitment | | | | | | | |
| Norway, no. (%) | 55 (53) | 84 (28) | 5 (8) | 65 (32) | 79 (31) | 28 (17) | 116 (39) |
| Denmark, no. (%) | 48 (47) | 211 (72) | 58 (92) | 141 (68) | 176 (69) | 136 (83) | 181 (61) |
| Recruitment site | | | | | | | |
| Physiotherapy, no. (%) | 30 (29) | 80 (27) | 25 (40) | 60 (29) | 75 (29) | 51 (31) | 84 (28) |
| Chiropractor, no. (%) | 42 (41) | 104 (35) | 14 (22) | 76 (37) | 84 (33) | 54 (33) | 106 (36) |
| General practice, no. (%) | 22 (21) | 45 (15) | 1 (2) | 26 (13) | 42 (16) | 13 (8) | 55 (19) |
| Outpatient clinic, no. (%) | 9 (9) | 66 (22) | 23 (37) | 44 (21) | 54 (21) | 46 (28) | 52 (17) |
| Age (years), mean (SD) | 27.4 (4.5) | 49.6 (8.3) | 70.4 (4.5) | 47.5 (15.2) | 47.6 (14.3) | 53.5 (14.0) | 44.2 (14.1) |
| Female, no. (%) | 56 (54) | 169 (57) | 30 (48) | 0 (0) | 255 (100) | 75 (46) | 180 (61) |
| Education, more than 12 years, no. (%) | 82 (80) | 192 (65) | 23 (37) | 117 (57) | 180 (71) | 0 (0) | 297 (100) |
| Family status, living with partner, no. (%) | 56 (54) | 105 (36) | 43 (69) | 86 (42) | 118 (46) | 78 (48) | 126 (42) |
| Employment, full-time, no. (%) | 66 (64) | 209 (71) | 5 (8) | 148 (72) | 132 (52) | 86 (52) | 194 (65) |
| Work characteristics*, sitting, no. (%) | 26 (33) | 133 (52) | 6 (46) | 78 (48) | 87 (48) | 29 (26) | 136 (58) |
| LBP, average last week, NRS (SD) | 4.8 (2.0) | 4.9 (1.9) | 5.0 (1.9) | 4.9 (1.9) | 4.9 (1.9) | 5.2 (1.8) | 4.7 (2.0) |
| LBP, worst pain last week, NRS (SD) | 6.5 (1.9) | 6.6 (2.0) | 6.6 (1.8) | 6.7 (1.8) | 6.5 (2.0) | 6.8 (1.9) | 6.5 (2.0) |
| LBP, duration more than 12 weeks, no. (%) | 52 (50) | 172 (58) | 43 (68) | 111 (54) | 156 (61) | 103 (63) | 164 (55) |
| LBP, every day within past year, no. (%) | 32 (31) | 126 (43) | 39 (62) | 85 (41) | 112 (44) | 80 (49) | 117 (39) |
| Daily use of pain medication, no. (%) | 53 (51) | 87 (29) | 17 (27) | 81 (39) | 76 (30) | 45 (27) | 112 (38) |

*Question only asked to those reporting full-or part time employment.

Abbreviations: no. = number, SD = Standard Deviation, LBP = low back pain, NRS = Numerical rating Scale

rates for answering the RMDQ were 80% at six-week, 87% at three-, 76% at six-, and 76% at nine-month follow-up.

Overall, we observed no effect modification of the sᴇʟꜰBACK intervention by age, gender, or education (Table 2). We observed no effect modification by gender or education at any time-point or for any outcomes (Tables 2–4). However, there was some evidence that age modified the effect at nine months follow-up; as the effect from sᴇʟꜰBACK was 4.7 RMDQ points (95% CI: 2.0 to 7.3) larger for participants ≥65 years as compared to participants aged 18–34 years, and 2.6 RMDQ points (95% CI: 0.4 to 4.9) larger in participants ≥65 years as compared to participants aged 35–64 years (Table 2).

Fig 1 shows the trajectories of LBP-related disability over the nine months follow-up period for (A) age, (B) gender, and (C) education strata. The number of participants in each stratum and mean values for the outcomes at all timepoints are given in S2–S4 Tables. For those aged ≥65 years, the control group had a different trajectory than all other age strata, i.e., they experienced an initial small reduction in LBP-related disability at six weeks and thereafter a regression towards the baseline level at nine months follow-up. In contrast, all other age strata experienced an initial reduction in LBP-related disability, which was sustained at three, six and nine months follow-up (Table 2 and Fig 1A).

**Table 2. Mean difference between groups and interaction effect at three and nine months for Roland Morris Disability Questionnaire.**

| | 3 months | | | 9 month | | |
|---|---|---|---|---|---|---|
| | Adjusted[a] mean difference (95% CI) SELFBACK vs usual care | Mean difference in effect due to interaction (95%CI) | | Adjusted[a] mean difference (95% CI) SELFBACK vs. usual care | Mean difference in effect due to interaction (95%CI) | |
| Age | | | | | | |
| 18–34 years | -0.8 (-2.4 to 0.8) | Reference | | 1.1 (-0.7 to 2.8) | Reference | |
| 35–64 years | -0.7 (-1.6 to 0.2) | 0.2 (-1.7 to 2.0) | Reference | -1.0 (-1.9 to -0.0) | -2.0 (-4.0 to -0.1) | Reference |
| ≥65 years | -1.7 (-3.7 to 0.3) | -0.8 (-3.4 to 1.7) | -1.0 (-3.2 to 1.2) | -3.6 (-5.7 to -1.6) | -4.7 (-7.3 to -2.0) | -2.6 (-4.9 to -0.4) |
| Gender | | | | | | |
| Female | -0.9 (-1.9 to 0.0) | Reference | | -0.7 (-1.7 to 0.34) | Reference | |
| Male | -0.6 (-1.6 to 0.5) | -0.4 (-1.8 to 1.1) | | -1.0 (-2.1 to 0.2) | 0.4 (-1.1 to 2.0) | |
| Education | | | | | | |
| ≤12 years | -1.1 (-2.3 to -0.1) | Reference | | -1.3 (-2.6 to -0.1) | Reference | |
| >12 years | -0.6 (-1.5 to 0.3) | 0.5 (-1.0 to 2.0) | | -0.6 (-1.5 to 0.4) | 0.7 (-0.8 to 2.3) | |

Abbreviations: UC = usual care, CI = Confidence interval

[a]Adjusted for stratification variables (recruitment site (physiotherapy, chiropractor, general practitioner, outpatient clinic), country (Norway, Denmark), duration of current LBP episode (>1, 1–4, 4–12, >12 weeks), LBP intensity (continuous, range 0–10), sex (male, female), age (continuous), education (<10, 10–12, >12 years). Note, when a variable is the dependent variable, it is not adjusted for in the analysis. A negative number indicate a further reduction in Roland-Morris Disability Questionnaire score for the SELFBACK group (intervention) compared to the usual care (control) group. For the mean difference in effect due to interaction, a positive number indicates a smaller reduction in LBP disability for the strata compared to the reference strata, a negative number indicates increased reduction in disability compared to the reference strata.

## Discussion

Age, gender, or education did not impact on the effect of the SELFBACK intervention on LBP-related disability. However, older participants may have an additional long-term positive effect compared to younger participants. The oldest group included only 63 participants, and thus the statistical uncertainty of the effect estimate within this subgroup is substantial. The difference in effect between the young and old age strata may be explained by differences in LBP characteristics. At baseline, more participants in the older age stratum compared to the 18–34 years age stratum reported that the current episode with LBP had lasted >12 weeks (68% vs. 50%), involved daily LBP (62% vs. 31%), but a lower daily consumption of pain medication (27% vs. 51%). This could indicate that participants between 18–34 years had more acute LBP, while those ≥65 years suffered from more chronic LBP. LBP characteristics might also affect the advice and treatment that the participants received from their clinicians (usual care). Monitoring of usual care was not a part of the RCT protocol, and a possible synergistic effect between self-management support and type of usual care for the different age groups cannot be excluded. However, another secondary analysis from the SELFBACK RCT did not find that the effect of the SELFBACK intervention was modified by the duration of the current LBP episode at baseline, nor by LBP intensity at baseline [31].

While several literature reviews have investigated the effect of digital interventions among elderly [32–35], we are not aware of any study directly comparing the effect of app-supported self-management between different age groups. The trajectory of LBP-related disability for those aged 18–34 years and 35–64 years was characterised by a steep reduction from baseline

**Table 3. Mean difference between groups and interaction effect at three and nine months for LBP intensity.**

| | 3 months | | | 9 months | | |
|---|---|---|---|---|---|---|
| | Adjusted[a] mean difference (95% CI) between selfBACK vs. usual care | Mean difference in effect due to interaction (95%CI) | | Adjusted[a] mean difference (95% CI) between selfBACK vs. usual care | Mean difference in effect due to interaction (95%CI) | |
| **Age** | | | | | | |
| 18–34 years | -0.7 (-1.4 to -0.1) | Reference | | -0.4 (-1.2 to 0.4) | Reference | |
| 35–64 years | -0.7 (-1.1 to -0.3) | -0.1 (-0.9 to 0.9) | Reference | -0.7 (-1.2 to -0.3) | -0.4 (-1.3 to 0.5) | Reference |
| ≥65 years | 0.1 (-0.8 to 1.1) | 0.8 (-0.4 to 2.0) | -0.8 (-0.2 to 1.9) | -0.9 (-1.8 to 0.1) | -0.5 (-1.7 to 0.7) | -0.1 (-1.2 to 0.9) |
| **Gender** | | | | | | |
| Female | -0.8 (-1.3 to -0.4) | Reference | | -0.6 (-1.1 to -0.1) | Reference | |
| Male | -0.4 (-0.9 to 0.1) | -0.4 (-1.1 to 0.3) | | -0.8 (-1.3 to -0.3) | 0.3 (-0.5 to 1.0) | |
| **Education** | | | | | | |
| ≤ 12 years | -0.7 (-1.3 to -0.1) | Reference | | -1.0 (-1.6 to -0.4) | Reference | |
| > 12 years | -0.6 (-1.0 to -0.2) | 0.1 (-0.6 to 0.9) | | -0.5 (-1.0 to -0.1) | 0.5 (-0.2 to 1.3) | |

Abbreviations: UC = usual care, CI = Confidence interval

[a]Adjusted for stratification variables (recruitment site (physiotherapy, chiropractor, general practitioner, outpatient clinic), country (Norway, Denmark), duration of current LBP episode (>1, 1–4, 4–12, >12 weeks), LBP intensity (continuous, range 0–10), sex (male, female), age (continuous), education (<10, 10–12, >12 years). Note, when a variable is the dependent variable, it is not adjusted for in the analysis. A negative number indicate a further reduction in LBP intensity score for the selfBACK group (intervention) compared to the usual care (control) group. For the mean difference in effect due to interaction, a positive number indicates a smaller reduction in pain intensity for the strata compared to the reference strata, a negative number indicates an larger reduction in pain intensity compared to the reference strata.

to six weeks follow-up, which thereafter was sustained over time (Fig 1A, Table 2). While a similar trajectory was seen for the intervention group in the ≥65 years stratum, the control group, on the other hand, had a small reduction from baseline to six-week follow-up, thereafter the mean score regressed towards the baseline value. A possible explanation of the different trajectory for the control group in the older age stratum may be that the effect of usual care differs across age strata. It may be hypothesized that with increasing age, participants become less mobile and able to self-transport to healthcare or that the growing need for healthcare with age results in a different prioritization of health issues when in contact with healthcare professionals. Using digital interventions, such as the selfBACK app, as a means for supporting disease management or remote interaction with healthcare professionals may be a way of overcoming this barrier [36]. The growing use of digital interventions is supported by emerging frameworks, such as from NICE (UK) [37], that aim to guide digital health interventions and technology development, to secure sustainable and trustworthy tools for the individual.

Regardless of the underlying mechanisms, this secondary analysis indicates that selfBACK is effective for reducing LBP-related disability across sociodemographic variables, also among older individuals with LBP. The results from the trial shows that older people are both willing and able to engage with digital interventions, which falls in line with emerging evidence for digital health interventions for elderly populations. Sohaib et al recently reported in a systematic review that mHealth interventions were effective for increasing physical activity or exercise in older people [33]. In a systematic review, Dunham et al found that older people are interested and willing to engage with mHealth apps that support chronic pain management[32]. Moreover, Parker et al found that most older people were willing to use mHealth apps for pain

**Table 4. Mean difference between groups and interaction effect at three and nine months for Pain Self-Efficacy Questionnaire.**

| | 3 months | | | 9 months | | |
|---|---|---|---|---|---|---|
| | Adjusted[a] mean difference (95% CI) between SELFBACK vs. usual care | Mean difference in effect due to interaction (95%CI) | | Adjusted[a] mean difference (95% CI) between SELFBACK vs. usual care | Mean difference in effect due to interaction (95%CI) | |
| **Age** | | | | | | |
| 18–34 years | 1.4 (-1.8 to 4.7) | Reference | | 0.6 (-2.9 to 4.0) | Reference | |
| 35–64 years | 2.9 (1.1 to 4.7) | 1.4 (-2.2 to 5.2) | Reference | 3.9 (2.0 to 5.8) | 3.3 (-6.4 to 7.2) | Reference |
| ≥65 years | 2.1 (-2.0 to 6.1) | 0.6 (-4.6 to 5.8) | -0.8 (-5.3 to 3.6) | 4.8 (0.6 to 8.9) | 4.2 (-1.2 to 9.6) | 0.9 (-3.6 to 5.5) |
| **Gender** | | | | | | |
| Female | 4.0 (2.0 to 5.9) | Reference | | 2.8 (0.7 to 4.8) | Reference | |
| Male | 0.7 (-1.5 to 2.9) | 3.2 (0.3 to 6.2) | | 3.8 (1.4 to 6.1) | -0.5 (-3.6 to 2.6) | |
| **Education** | | | | | | |
| ≤ 12 years | 3.7 (1.2 to 6.2) | Reference | | 3.4 (0.8 to 6.0) | Reference | |
| > 12 years | 1.9 (0.1 to 3.7) | -1.8 (-4.9 to 1.3) | | 3.2 (1.3 to 5.1) | -0.2 (-3.5 to 3.0) | |

Abbreviations: UC = usual care, CI = Confidence interval

[a]Adjusted for stratification variables (recruitment site (physiotherapy, chiropractor, general practitioner, outpatient clinic), country (Norway, Denmark), duration of current LBP episode (>1, 1–4, 4–12, >12 weeks), LBP intensity (continuous, range 0–10), sex (male, female), age (continuous), education (<10, 10–12, >12 years). Note, when a variable is the dependent variable, it is not adjusted for in the analysis. A negative number indicate a further reduction in Pain Self-Efficacy score for the SELFBACK group (intervention) compared to the usual care (control) group. For the mean difference in effect due to interaction, a positive number indicates a larger reduction in pain self-efficacy for the strata compared to the reference strata, a negative number indicates an smaller reduction in pain self-efficacy compared to the reference strata.

management and that tailoring of equipment (e.g., wearables) and content facilitate use [16]; these tailoring factors were utilised in the SELFBACK intervention.

We previously conducted a systematic review on digital support interventions for self-management of LBP and found that the included population was predominantly female (59–83% in the six included RCTs) [14]. Since this review in 2016, similar uneven gender distribution has been seen in other trials on digital interventions for LBP management [38,39] as well as in the SELFBACK RCT [11]. This secondary analysis did not find evidence to support effect modification by gender, and we have not identified any other studies investigating this aspect.

A previous study found that educational level predicted ability to self-manage chronic LBP [17], and educational level is argued to affect engagement and acceptability of digital interventions [18,19]. Rabenbauer et al examined factors associated with effectiveness of eHealth interventions for chronic LBP and proposed a mediating role of self-efficacy for creating healthy habits and that eHealth literacy is associated with self-efficacy in this model [40]. They found that education acted as a confounder or modifier of self-efficacy, thereby suggesting an influence on the individual's ability to create healthy habits. This trial did not find evidence supporting effect modification by educational level. All SELFBACK content was designed with input from a patient user group, which is likely to have helped with the accessibility and readability of the content for all education levels. A process evaluation has been conducted concurrently to the RCT [41]. The results from the process evaluation will inform the further development of mHealth interventions in general and provide valuable knowledge to be utilized in the personalization and individual tailoring of the SELFBACK intervention.

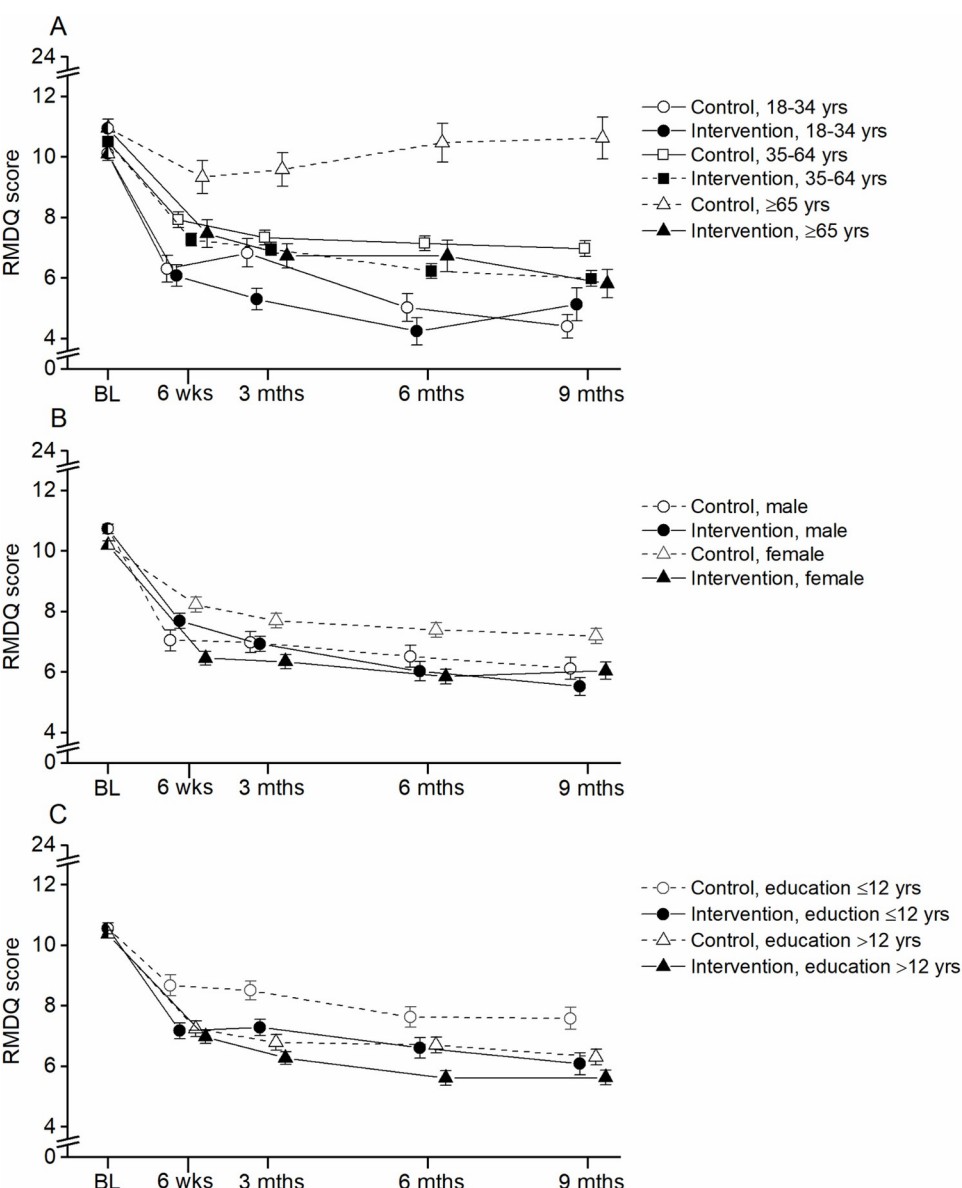

**Fig 1.** Trajectory of RMDQ scores at all time points for participants stratified by A) age, B) gender, and C) education. RMDQ: Roland-Morris Disability Questionnaire.

Strengths of the current study include high external validity as the included population is heterogenous and recruited from primary care and from an outpatient clinic. The selfBACK RCT included participants across a wide age-range (18–86 years), complementing previous studies, which mainly included middle-aged adults [14]. This heterogeneity has enabled several secondary analyses investigating effect modification across LBP characteristics, such as multimorbidity and co-occurring musculoskeletal pain [13] and pain intensity and duration at baseline [31]. Further, secondary analyses may identify subgroups more likely to benefit from the intervention than other subgroups; however, we have so far not found any consistent effect modification related to the selfBACK intervention, which may be due to the individual tailoring of the intervention [10]. Nevertheless, future studies are required to assess whether AI-based tailoring introduce additional beneficial effects compared to non-tailored digital

support. In addition, a number of studies are ongoing to inform the further development of the app, these include results from the parallel process evaluation and perspectives of cost-effectiveness.

A methodological consideration for this secondary analysis was the choice of outcomes as well as the chosen cut-points for stratification. The outcomes, LBP-related disability, LBP intensity and pain self-efficacy were chosen as they are indicative of core outcomes for LBP trials [27]. Also, together these outcomes cover the main focus of self-management trials, i.e. manage symptoms and make choices that may improve wellbeing, function and pain level [8]. The choice of cut-points for stratifying the potential effect modifiers was informed by literature as described in the introduction. However, the specific cut-point may affect the comparison of results to other studies applying different outcomes or cut-points. It will also affect the number of participants in each stratum, potentially resulting in an unbalanced allocation between intervention and control group in the different strata [42]. Furthermore, it should be noted that although these secondary analyses were a priori planned for, the sample size of the SELFBACK RCT was not powered for subgroup analyses. Similarly, additional secondary analysis investigating effect modification of the trial has been performed. This taken together there is likelihood of chance findings in these secondary analyses. Consequently, the results on effect modification should be interpreted with caution [43].

## Conclusion

Age, gender, or education did not impact on the effect of the SELFBACK intervention on LBP-related disability. However, older participants may have an additional long-term positive effect compared to younger participants. This suggests that the SELFBACK intervention may benefit all persons seeking care for LBP in primary care regardless of age, gender, and education, and be a helpful tool for clinicians and patients to support self-management of LBP.

## Supporting information

**S1 CONSORT Checklist. CONSORT checklist for the secondary analysis on effect modification from age, gender and educational level.**
(DOC)

**S1 Table. Population characteristics across sociodemographic variables for intervention and control group.**
(DOCX)

**S2 Table. Mean and difference between groups at three and nine months for Roland Morris Disability Questionnaire.**
(DOCX)

**S3 Table. Mean and difference between groups at three and nine months for LBP intensity.**
(DOCX)

**S4 Table. Mean and difference between groups at three and nine months for Pain Self-Efficacy Questionnaire.**
(DOCX)

## Author Contributions

**Conceptualization:** Ellen Marie Bardal, Louise Fleng Sandal, Tom Ivar Lund Nilsen, Barbara I. Nicholl, Paul Jarle Mork, Karen Søgaard.

**Data curation:** Ellen Marie Bardal.

**Formal analysis:** Louise Fleng Sandal, Tom Ivar Lund Nilsen.

**Funding acquisition:** Tom Ivar Lund Nilsen, Barbara I. Nicholl, Paul Jarle Mork, Karen Søgaard.

**Investigation:** Paul Jarle Mork, Karen Søgaard.

**Methodology:** Louise Fleng Sandal, Tom Ivar Lund Nilsen, Barbara I. Nicholl.

**Project administration:** Barbara I. Nicholl, Paul Jarle Mork, Karen Søgaard.

**Supervision:** Paul Jarle Mork, Karen Søgaard.

**Visualization:** Ellen Marie Bardal.

**Writing – original draft:** Ellen Marie Bardal, Louise Fleng Sandal, Paul Jarle Mork.

**Writing – review & editing:** Tom Ivar Lund Nilsen, Barbara I. Nicholl, Karen Søgaard.

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
