## [Decision Letter · Decision Letter 0]

8 Feb 2023

PDIG-D-22-00340

Evaluation of effectiveness of the AI-based selfBACK app for low back pain across sociodemographic groups: Secondary analysis of the selfBACK randomised controlled trial

PLOS Digital Health

Dear Dr. Bardal,

Thank you for submitting your manuscript to PLOS Digital Health. After careful consideration, we feel that it has merit but does not fully meet PLOS Digital Health's publication criteria as it currently stands. Therefore, we invite you to submit a revised version of the manuscript that addresses the points raised during the review process.

Please submit your revised manuscript within 60 days Apr 09 2023 11:59PM. If you will need more time than this to complete your revisions, please reply to this message or contact the journal office at digitalhealth@plos.org. Please include the following items when submitting your revised manuscript:

We look forward to receiving your revised manuscript.

Kind regards,

Crina Grosan

Academic Editor

PLOS Digital Health

Journal Requirements:

Additional Editor Comments (if provided):

Reviewers' comments:

Reviewer's Responses to Questions

**Comments to the Author**

1. Does this manuscript meet PLOS Digital Health’s publication criteria? Is the manuscript technically sound, and do the data support the conclusions? The manuscript must describe methodologically and ethically rigorous research with conclusions that are appropriately drawn based on the data presented.

Reviewer #1: Partly

Reviewer #2: Yes

Reviewer #3: Yes

2. Has the statistical analysis been performed appropriately and rigorously?

Reviewer #1: Yes

Reviewer #2: Yes

Reviewer #3: Yes

3. Have the authors made all data underlying the findings in their manuscript fully available (please refer to the Data Availability Statement at the start of the manuscript PDF file)?

Reviewer #1: No

Reviewer #2: Yes

Reviewer #3: Yes

4. Is the manuscript presented in an intelligible fashion and written in standard English?

Reviewer #1: Yes

Reviewer #2: Yes

Reviewer #3: Yes

5. Review Comments to the Author

Reviewer #1: <Main comments>

The authors examined the effect modification of the promising digital health intervention app, the selfBACK, and clearly showed that pain-related disability was not modified by age, gender, education. However, this work still contains some inadequacies in the data analysis and discussions. Although I understand that it is unrealistic to acquire new clinical data because this study is a secondary analysis of the previous randomized controlled trial, I listed some issues that need to be addressed in order to improve this work mature enough for publication in this journal.

1. Figure 1A and table 2 indicate that there is a tendency for the selfBACK intervention to be more effective in the elderly and less effective in the rest of the population, even though without any significance. Could it be a reason for the results of the RCT that is mentioned as “the effect was as small and of uncertain clinical significance” at line 66? The authors should discuss this possibility to try to explain about the uncertainty in the RCT.

2. Do the authors have the detailed information about what kind of usual care the participants took at each time point of the RCT? If they do, they should clearly show the information for both the intervention group and the control group to help readers further understand the effect modification of the selfBACK.

3. This work including the previous RCT does not seem to have shown how AI-based tailored support improved LBP, because it might be still possible that the usage of a digital tool without AI in itself could encourage the LBP self-management enough. Nevertheless, at line 250-251, the authors mentioned “we have so far not found any consistent effect modification related to the selfBACK intervention, which may be due to the individual tailoring of the intervention.” This sentence could lead to a biased discussion. If the authors use the word “AI-based” in the title of this article, they should show or discuss whether it is the AI-based tailored support or just a support by a digital tool that improved the LBP self-management.

4. The author should clarify what kind of future research plans they need to proceed to based on the results of this secondary analysis, which readers might strongly want to know for further development of this promising app.

<Minor points>

I listed below some points which might be typos. Please correct each of them, if it should be corrected.

1. Both “SELFBACK” and “selfBACK” can be seen through the whole texts. Which is correct?

2. At line 167: “35-64 years” - a typo for “18-34 years” ?

3. In table 2: With mean difference between genders for RMDQ, 

 “-0.4” at 3 months - “0.4” ? 

 “0.4” at 9 months - “-0.4” ? 

4. In table 3: With mean difference between genders for LBP intensity, 

 “-0.4” at 3 months - “0.4” ? 

 “0.3” at 9 months - “-0.3” ?

5. In table 4: With mean difference between genders for Pain Self-Efficacy Questionnaire, 

 “3.2” at 3 months - “-3.2” ? 

 “-0.5” at 9 months - “1.0” ?

Reviewer #2: The manuscript is technically sound. The objective is well framed. The methodology to address the objective is well covered. Explanation has been given of how the study participants were recruited. The study design (RCT) is appropriate as the interest is on the treatment effect . Ethical consideration has been captured. The conclusions are based on the findings from the study using the data that was collected.

The statistical analysis is appropriate though a linear model with the treatment, age, gender and education as independent variables could have been explored.

All the data have been made available. 

The manuscript is well structured and written in standard English

Reviewer #3: Overall, the manuscript is very well written and clear. I only have some very minor comments:

1) Some sentences could be added in the introduction section on how this work extends the previous works on this app by the authors. The contributions of this work compared to the four past papers by the team could be elaborated.

2) How does lower back pain monitoring impact workplace productivity? Is there any study or future scope to study how these participants manage their pain while at the workplace? This could be discussed in the discussion section, for example, on the use of your app outdoors or at work. There are some recent apps and devices that are applicable for self-monitoring at the workplace. A review paper is here for reference: https://doi.org/10.1002/aisy.202100099, V.B. Patel et al, Advanced Intelligent Systems, 4: 2100099, 2022

6. PLOS authors have the option to publish the peer review history of their article (what does this mean?). If published, this will include your full peer review and any attached files.

**Do you want your identity to be public for this peer review?** For information about this choice, including consent withdrawal, please see our Privacy Policy.

Reviewer #1: No

Reviewer #2: No

Reviewer #3: No

---

## [Decision Letter · Decision Letter 1]

19 Jun 2023

Do age, gender, and education modify the effectiveness of app-delivered and tailored self-management support among adults with low back pain? -Secondary analysis of the selfBACK randomised controlled trial

PDIG-D-22-00340R1

Dear Dr. Bardal,

We are pleased to inform you that your manuscript 'Do age, gender, and education modify the effectiveness of app-delivered and tailored self-management support among adults with low back pain? -Secondary analysis of the selfBACK randomised controlled trial' has been provisionally accepted for publication in PLOS Digital Health.

Best regards,

Crina Grosan

Academic Editor

PLOS Digital Health

Reviewer Comments (if any, and for reference):

Reviewer's Responses to Questions

**Comments to the Author**

1. If the authors have adequately addressed your comments raised in a previous round of review and you feel that this manuscript is now acceptable for publication, you may indicate that here to bypass the “Comments to the Author” section, enter your conflict of interest statement in the “Confidential to Editor” section, and submit your "Accept" recommendation.

Reviewer #1: All comments have been addressed

Reviewer #2: All comments have been addressed

Reviewer #3: All comments have been addressed

2. Does this manuscript meet PLOS Digital Health’s publication criteria? Is the manuscript technically sound, and do the data support the conclusions? The manuscript must describe methodologically and ethically rigorous research with conclusions that are appropriately drawn based on the data presented.

Reviewer #1: Yes

Reviewer #2: Yes

Reviewer #3: Yes

3. Has the statistical analysis been performed appropriately and rigorously?

Reviewer #1: Yes

Reviewer #2: Yes

Reviewer #3: Yes

4. Have the authors made all data underlying the findings in their manuscript fully available (please refer to the Data Availability Statement at the start of the manuscript PDF file)?

Reviewer #1: Yes

Reviewer #2: Yes

Reviewer #3: Yes

5. Is the manuscript presented in an intelligible fashion and written in standard English?

Reviewer #1: Yes

Reviewer #2: Yes

Reviewer #3: Yes

6. Review Comments to the Author

Reviewer #1: The authors have addressed all the comments to the best of their available data within the constraints as a secondary analysis. I have no further comments at this time. Their current work is a promising technology to resolve worldwide health problems of low back pain and I really look forward to further development of their work.

Reviewer #2: Thanks a lot to the authours for their time in clarifying and addressing the issues raised by the reviewers.

The Title of the article is now clear.

The issue of sample size determination has been clarified.

Minor Suggestion

the authours have made the following statement on Lines 46-47, 60-61, 211-212, 302-303

"However, older participants may have an additional long term positive effect compared to younger participants"

I suggest that the statement be replaced by

"However, the SELFBACK intervention may have an additional long term positive effect on the older participants compared to younger participants"

Line 97: "modifies" should read "modifiers"

Line 239: "use of digital intervention in is supported...." should read "use of digital intervention is supported...."

The abbreviation UC - usual care needs to be captured in the List of abbreviations

Reviewer #3: The authors have addressed all the questions from the reviewers satisfactorily.

7. PLOS authors have the option to publish the peer review history of their article (what does this mean?). If published, this will include your full peer review and any attached files.

**Do you want your identity to be public for this peer review?** For information about this choice, including consent withdrawal, please see our Privacy Policy.

Reviewer #1: No

Reviewer #2: No

Reviewer #3: **Yes: **Santosh Pandey
